# Synthesis and versatile reactivity of scandium phosphinophosphinidene complexes

Bin Feng[1], Li Xiang[1], Karl N. McCabe[2], Laurent Maron [2✉], Xuebing Leng[1] & Yaofeng Chen [1✉]

M=E/M≡E multiple bonds (M = transition metal, E = main group element) are of significant fundamental scientific importance and have widespread applications. Expanding the ranges of M and E represents grand challenges for synthetic chemists and will bring new horizons for the chemistry. There have been reports of M=E/M≡E multiple bonds for the majority of the transition metals, and even some actinide metals. In stark contrast, as the largest subgroup in the periodic table, rare-earth metals (Ln) were scarcely involved in Ln=E/Ln≡E multiple bonds. Until recently, there were a few examples of rare-earth monometallic alkylidene, imido and oxo complexes, featuring Ln=C/N/O bonds. What are in absence are rare-earth monometallic phosphinidene complexes with Ln=P bonds. Herein, we report synthesis and structure of rare-earth monometallic phosphinidene complexes, namely scandium phosphinophosphinidene complexes. Reactivity of scandium phosphinophosphinidene complexes is also mapped out, and appears to be easily tuned by the supporting ligand.

[1] State Key Laboratory of Organometallic Chemistry, Shanghai Institute of Organic Chemistry, University of Chinese Academy of Sciences, Chinese Academy of Sciences, Shanghai, PR China. [2] LPCNO, CNRS & INSA, Université Paul Sabatier, Toulouse, France. ✉email: laurent.maron@irsamc.ups-tlse.fr; yaofchen@mail.sioc.ac.cn

The synthesis and reactivity of M=E/M≡E multiple bonds (M = transition metal, E = main group element) is one of the most vibrant areas of modern chemistry. The research in this area stimulated fundamental development of chemistry i.e., chemical bonding theory, and found widespread applications in organic and polymer synthesis[1–3]. The most notable example is the olefin metathesis based on the M=C bonds, for which the Nobel Prize for Chemistry was awarded in 2005[4–6]. The reactivity profile of the M=E/M≡E multiple bonds largely depends on the interaction between the $d$ orbitals of metal and the $p$ orbitals of ligand E atom. Insufficient overlap creates an electronically frustrated moiety, thereby providing the electrophilic or nucleophilic metal center or ligand E atom[7].

There have been reports of M=E/M≡E interactions for the majority of the metallic elements of the periodic table, even some actinide metals. In stark contrast, the largest subgroup of the periodic table, rare-earth metals (Ln: Sc, Y, and lanthanides), was scarcely involved in metal–ligand multiple bonds. The scarcity is mainly attributed to energy mismatching between the frontier orbitals of the rare-earth metals and the ligand atoms. This renders the putative Ln=E/Ln≡E bonds extremely instable, which are readily labile to aggregation and/or reaction with the ligand/environment, quenching the multiple-bond character[7–9]. However, the other side of the coin is, if the extremely high reactivity can be tamed, the Ln=E/Ln≡E species will lead to a vast new regime of novel reactivity. But realization of the prospect is hampered by the limited access to the Ln=E/Ln≡E species, only a few structurally characterized pincer-type rare-earth monometallic alkylidene complexes had been reported before 2010[10,11]. Since the year of 2010, the landscape in the area changed gradually, as a few rare-earth monometallic imido[12–14] and oxo complexes were reported[15,16]. However, a significant knowledge gap in the area is rare-earth monometallic phosphinidene, the efforts in this field only led to some rare-earth bi- or trimetallic phosphinidene complexes[17–22] or Li/Sc heterobimetallic phosphinidene complexes[23]. This is not unexpected, as the rare-earth metal ions are among the hardest Lewis acids, while the phosphorus atom is the softest Lewis base, the bonding between rare-earth metal and phosphorus is generally weaker than that between rare-earth metal and nitrogen or oxygen. To tackle the challenge, a well-designed phosphinidene ligand is essential.

In this work, by introducing a phosphinophosphinidene, [PP{N(DIPP)CH$_2$CH$_2$N(DIPP)}]$^{2-}$, into rare-earth chemistry, we are able to synthesize scandium phosphinophosphinidene complexes, which have the monometallic structure. The reactivity of scandium phosphinophosphinidene complexes is also studied and appears to be easily tuned by the supporting ligand. Particularly, in one complex, the reactivity occurs at the least nucleophilic phosphorus of the phosphinophosphinidene ligand. This is rationalized by computational approaches and is due to the Lewis acidity of the metal center that binds tightly THF molecule.

## Results

**Synthesis and structural characterization.** Scandium methyl chlorides [LSc(Me)Cl] (L = [MeC(NDIPP)CHC(NDIPP)Me]$^-$, DIPP = 2,6-($^i$Pr)$_2$C$_6$H$_3$)[24], and [L'Sc(Me)Cl] (L' = [MeC(NDIPP)CHC(Me)(NCH$_2$CH$_2$N(Me)$_2$)]$^-$)[25] were prepared as reported. Phosphinophosphine H$_2$PP{N(DIPP)CH$_2$CH$_2$N(DIPP)} was synthesized by a salt metathesis of ClP{N(DIPP)CH$_2$CH$_2$N(DIPP)}[26,27] with NaPH$_2$ in THF. The $^{31}$P NMR spectrum of the compound in C$_6$D$_6$ exhibits two sets of doublet-triplets at $\delta$ = −157.6 ($dt$, $^1J_{P-P}$ = 186 Hz, $^1J_{P-H}$ = 183 Hz, P$_\alpha$) and 134.9 ppm ($dt$, $^1J_{P-P}$ = 186 Hz, $^2J_{P-H}$ = 16 Hz, P$_\beta$). This compound was also characterized by single crystal X-ray diffraction (XRD) (Supplementary Fig. 1). The P–P bond length in the compound is 2.277

(1) Å, in line with a P–P single bond. This phosphinophosphine can be deprotonated by KCH$_2$(C$_6$H$_5$) in THF to give a potassium salt K[HPP{N(DIPP)CH$_2$CH$_2$N(DIPP)}], which is stable in THF but decomposes when the THF is removed. Therefore the in situ generated K[HPP{N(DIPP)CH$_2$CH$_2$N(DIPP)}] was treated with [LSc(Me)Cl] or [L'Sc(Me)Cl] in a THF/toluene mixture, and the reactions afforded scandium phosphinophosphinidene complexes **1** and **2** in 54% and 67% yields, respectively, (Fig. 1). In the $^{31}$P {$^1$H} NMR spectra of **1** and **2** in THF-$d_8$, the P$_\alpha$(phosphinidene) signals appear at $\delta$ = 402.3 ppm ($d$, $^1J_{P-P}$ = 501 Hz) and 312.2 ($d$, $^1J_{P-P}$ = 519 Hz), respectively, which are dramatically downshifted in comparison with the P$_\alpha$(phosphido) signal in the $^{31}$P NMR spectrum of the potassium salt K[HP{PN(DIPP)CH$_2$CH$_2$N (DIPP)}] (−119.0 ppm, $dd$, $^1J_{P-P}$ = 423 Hz, $^1J_{P-H}$ = 128 Hz). Compared with that of the phosphinophosphine, the P–P coupling constants for either the potassium salt or the scandium phosphinophosphinidene complexes are significantly larger; this indicates a shorter P–P bond and a delocalization of the negative charge of the P$_\alpha$(phosphido) or P$_\alpha$(phosphinidene) atom into the P$_\beta$(phosphino) atom.

The XRD studies on the single crystals of **1** and **2** show both complexes contain a $^2\eta$-bonded phosphinophosphinidene ligand (Fig. 2). In **1**, the Sc–P$_\alpha$ bond length (2.448(1) Å) (Table 1) is shorter than those found in scandium bridged phosphinidene complexes, [(LSc)$_2${$\mu_2$-CH$_2$}{$\mu_2$-P(DIPP)}] (2.495(1) and 2.508(1) Å)[22], and [{(MeC(NDIPP)CHC(Me)(NCH$_2$CH$_2$N($^i$Pr)$_2$)Sc}$_2${$\mu_2$-P(DIPP)}$_2$] (2.522(1) and 2.528(1) Å)[20]. This is on the other hand longer than that in a lithium capped scandium phosphinidene ate complex [(PNP)Sc{$\mu_2$-P(C$_6$H$_3$-2,6-Mes$_2$)}($\mu_2$-Br)Li] (2.338(2) Å)[23]. The Sc–P$_\beta$ bond length (2.718(1) Å) is much longer than the Sc–P$_\alpha$ bond length (2.448(1) Å) in the complex, and also longer than the Sc–P single bond lengths in a scandium diphosphido complex [LSc{PH(DIPP)}$_2$] (2.570(3) and 2.609(3) Å)[22]. The P$_\alpha$–P$_\beta$ bond length in **1**, 2.105(1) Å, is shorter than that observed in the phosphinophosphine (2.277(1) Å). This is in line with the observed larger P–P coupling constant for **1** compared with that for the phosphinophosphine. The Sc–P$_\alpha$ and Sc–P$_\beta$ bond lengths in **2** are longer than those in **1**, 2.484(1) and 2.814(1) Å vs 2.448(1) and 2.718(1) Å, due to an increasing in the coordination number of scandium from five

**Fig. 1 Synthesis of scandium phosphinophosphinidene complexes.** Deprotonation of phosphinophosphine H$_2$PP{N(DIPP)CH$_2$CH$_2$N(DIPP)} with KCH$_2$(C$_6$H$_5$) in THF gives a potassium salt K[HPP{N(DIPP)CH$_2$CH$_2$N (DIPP)}], which subsequently reacts with scandium methyl chlorides [LSc (Me)Cl] or [L'Sc(Me)Cl] to give scandium phosphinophosphinidene complexes **1** and **2**.

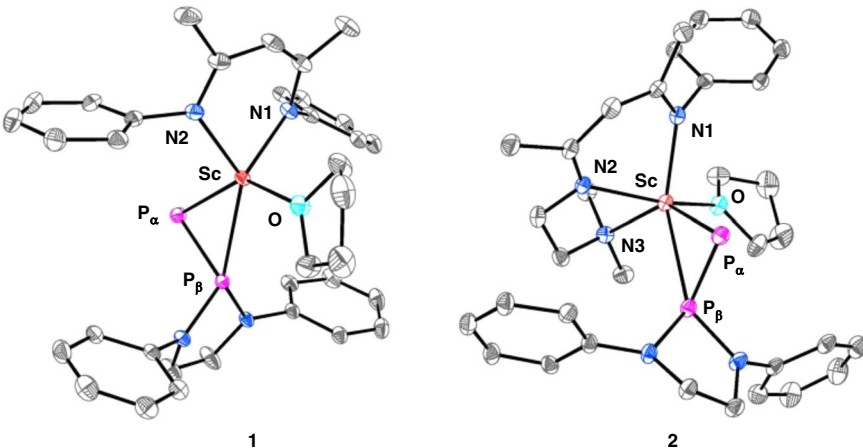

**Fig. 2 Molecular structures of complexes 1 and 2 with ellipsoids at 30% probability level.** DIPP isopropyl groups and hydrogen atoms were omitted for clarity.

**Table 1 Important bond lengths/angles of complexes 1–9, scandium bridged phosphinidene complex [(LSc)$_2$($\mu_2$-CH$_2$){$\mu_2$-P(DIPP)}] (A), lithium capped scandium phosphinidene ate complex [(PNP)Sc{$\mu_2$-P(C$_6$H$_3$-2,6-Mes$_2$)}($\mu_2$-Br)Li] (B) and scandium diphosphido complex [LSc{PH(DIPP)}$_2$] (C).$^a$**

| Entry | 1 | 2 | 3 | 4 | 5 | 6 | 7 | 8 | 9 | A | B | C |
|---|---|---|---|---|---|---|---|---|---|---|---|---|
| Sc–P$_\alpha$ | 2.448(1) | 2.484(1) | 2.547(1) | 2.557(1) | 2.535(2) | 2.505(1) | 2.508(1) | 2.618(1) | 2.544(1) | 2.502(1)$^b$ | 2.338(2) | 2.590(3)$^b$ |
| Sc–P$_\beta$ | 2.718(1) | 2.814(1) | – | – | – | – | – | – | – | – | – | – |
| P$_\alpha$–P$_\beta$ | 2.105(1) | 2.095(1) | 2.211(1) | 2.027(1) | 2.021(1) | 2.097(2) | 2.050(1) | 2.229(1) | 2.222(1) | – | – | – |
| Sc–P$_\alpha$–P$_\beta$ | 72.9(1) | 75.3(1) | 121.3(1) | 96.7(1) | 89.7(1) | 91.1(1) | 92.4(1) | 111.6(1) | 110.8(1) | – | – | – |

$^a$Bond lengths [Å] and bond angles [°].
$^b$The average value of two Sc–P bond lengths in the complex.

to six. The P$_\alpha$–P$_\beta$ bond length in **2**, 2.095(1) Å, is similar to that in **1**, 2.105(1) Å.

**Reactivity.** As expected for a Sc–P$_\alpha$ multiple bond, complex **1** reacts with N-benzylidenepropylamine at room temperature to give a [2 + 2] cycloaddition product **3** (Fig. 3). This reactivity contrasts with one observed for the coordination-free phosphinophosphinidenes, which are electrophilic[28]. In **3** (Fig. 4), the P$_\beta$ atom is not coordinated to the scandium center. The Sc–P$_\alpha$ bond length is longer than that in **1**, 2.547(1) vs 2.448(1) Å, which is in accordance with the decrease of the bond order. The P–P bond length (2.211(1) Å) is longer than in **1** (2.105(1) Å) but shorter than that in the phosphinophosphine (2.277(1) Å). Accordingly, in the $^{31}$P{$^1$H} NMR spectra in C$_6$D$_6$, the P–P coupling constant for **3** (384 Hz) is smaller than that for **1** (505 Hz) but larger than that for the phosphinophosphine (186 Hz). The chemical shifts of the P atoms for **3** are also dramatically changed, the P$_\alpha$ signal appears at $\delta = 40.8$ ppm and the P$_\beta$ signal is at $\delta = 169.0$ ppm.

In attempts to synthesize the scandium end-on phosphinophosphinidene, the addition of a strong donor, namely the bipyridine (bpy), to **1** was investigated. Surprisingly, the reaction gives a bpy-insertion product **4** instead of a bpy-coordination compound in a nearly quantitative yield. During the reaction, the P$_\beta$ atom of **1** nucleophilically attacks one *ortho*-carbon atom of bpy. This is a unique reactivity, as in all previously reported phosphinophosphinidene metal complexes, the P$_\beta$ atom hardly reacts as a nucleophile with unsaturated substrates[29]. The newly formed $sp^3$ carbon atom in **4** displays a featured signal at $\delta = 78.3$ ppm (dd, $^1J_{P-C} = 77.7$ Hz, $^2J_{P-C} = 9.3$ Hz) in its $^{13}$C{$^1$H} NMR spectrum. In the $^{31}$P{$^1$H} NMR spectrum of the complex, the P$_\alpha$ and P$_\beta$ signals appear at $\delta = -64.4$ and 110.0 ppm, respectively; the P–P coupling constant is up to 639 Hz. In constant with a large P–P

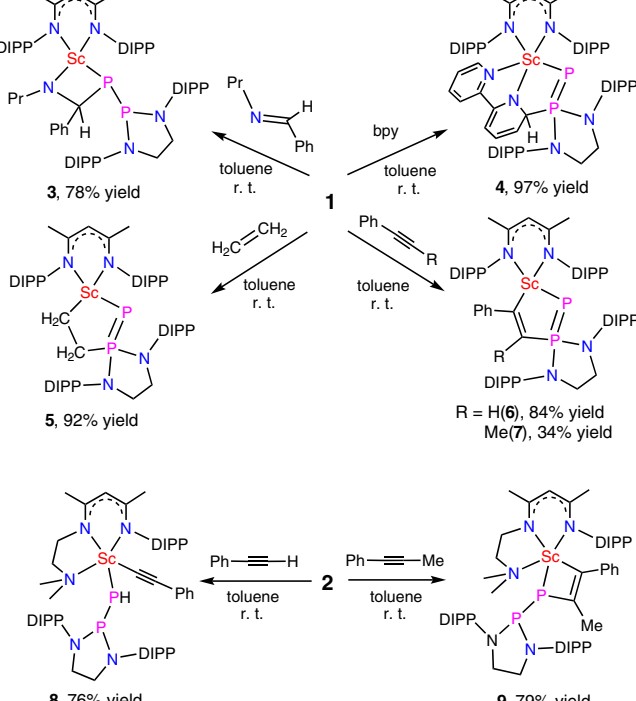

**Fig. 3 Reactivity of scandium phosphinophosphinidene complexes.** Reactions of complex **1** with N-benzylidenepropylamine, 2,2′-bipyridine, ethylene, phenylacetylene and 1-phenyl-1-propyne, and reactions of complex **2** with phenylacetylene and 1-phenyl-1-propyne.

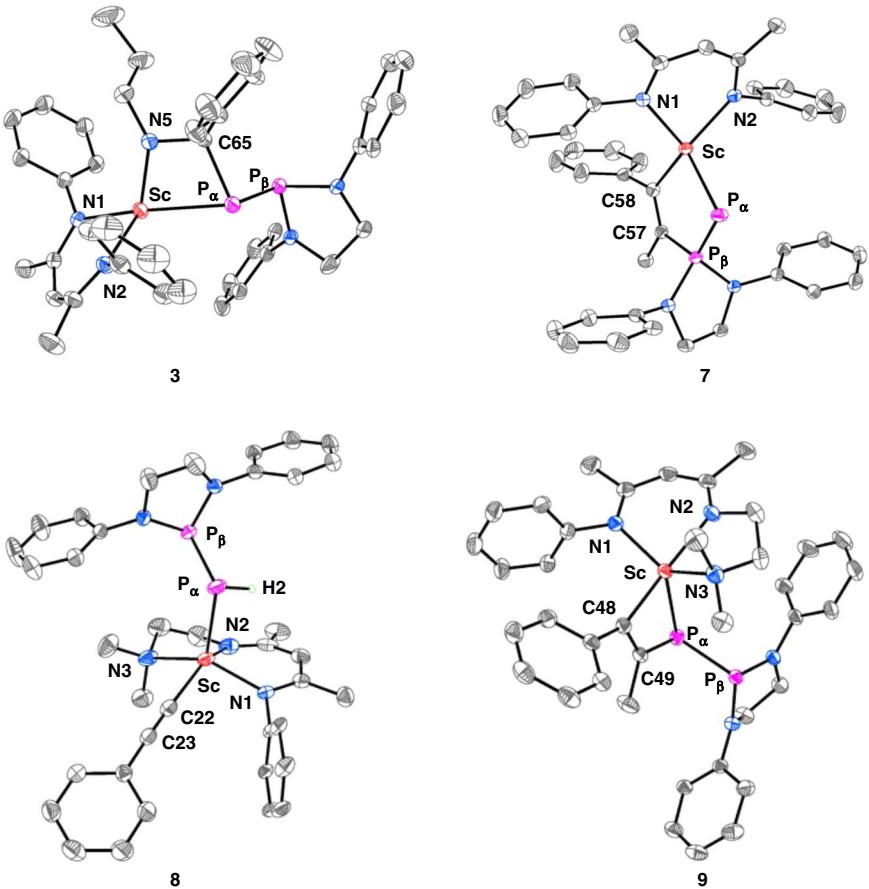

**Fig. 4 Molecular structures of complexes 3 and 7–9.** The ellipsoids of **3** are at 30% probability level, while those of **7**, **8**, and **9** are at 40% probability level. DIPP isopropyl groups and hydrogen atoms (except the hydrogen atom on $P_\alpha$ in **8**) were omitted for clarity.

coupling constant observed in the solution NMR study, the XRD studies on the single crystal of **4** indicate the complex has a short P–P bond length, 2.027(1) Å (Supplementary Fig. 5), which is similar to the P–P double bond lengths of the diphosphenes (~2.03 Å)[30]. As expected, the $Sc–P_\alpha$ bond length in **4** (2.557(1) Å) is longer than that in **1** (2.448(1) Å), but close to that in **3** (2.547 (1) Å).

Complex **1** also reacts with ethylene and phenylacetylene, yielding an insertion of the C–C double or triple bond into the $Sc–P_\beta$ bond (instead of the $Sc–P_\alpha$ bond), and the products **5** and **6** were isolated in high yields (Fig. 3). The reaction with phenylacetylene is highly regioselective, only the isomer **6** which minimizes the steric repulsion between the phenyl group of the alkyne and the DIPP group of the phosphinophosphinidene was obtained. The electronic effects of the phenyl substituent on the reaction also favor the formation of this isomer as it stabilizes the partial negative charge on the benzylic carbon when it is located α to the metal ion. The reaction of **1** with 1-phenyl-1-propyne gives several products, complex **7** (Fig. 3) was isolated in 34% yield while other products could not be isolated. The complex **7** comes from the insertion of the C–C triple bond of 1-phenyl-1-propyne into the $Sc–P_\beta$ bond of **1**, with the phenyl substituent located α to metal ion. Similar to that of **4**, the $^{31}P\{^1H\}$ NMR spectra of **5–7** in solutions show the large P–P coupling constants, 634, 615, and 647 Hz, respectively. In the solid states, the P–P bond lengths of **5**, **6** (Supplementary Figs. 5, 6), and **7** (Fig. 4) are 2.021(2), 2.097(2), and 2.050(1) Å, respectively. The newly formed Sc–C and P–C bonds in **5**, **6**, and **7** are 2.126(5) and 1.840(5) Å, 2.253(4) and 1.797(4) Å, and 2.234(2) and 1.844(2) Å, respectively. It is worthy to note that the reactions occur on the $Sc–P_\beta$ bond of **1** are similar

to those observed for some metal-based frustrated Lewis pairs, where the unsaturated substrates inserted into the metal–phosphorus functions[31–34].

In contrast to the reactivity of **1**, complex **2** reacts with phenylacetylene to give a scandium phenylacetylide **8** (Fig. 3). The $P_\alpha$ of **2** abstracts a proton from phenylacetylene in the reaction, and this resembles the reactivity of a scandium imido complex[35]. The single crystals of **8** were obtained and characterized by XRD (Fig. 4). The $Sc–P_\alpha$ and P–P bond lengths in **8** are both significantly longer than those in **2**, 2.618(1) and 2.229(1) Å vs 2.484(1) and 2.095(1) Å. Furthermore, complex **2** nearly quantitatively undergoes a [2 + 2] cycloaddition with 1-phenyl-1-propyne, and complex **9** is isolated in 79% yield. In the $^{31}P\{^1H\}$ NMR spectra, the $P_\alpha$ and $P_\beta$ signals of **9** appear at $\delta =$ 70.6 and 173.5 ppm, which are significantly downshifted in comparison with those of **7** (−18.6 and 80.3 ppm); the P–P coupling constant for **9** is much smaller than that for **7**, 361 Hz vs 647 Hz. Complex **9** was also characterized by XRD (Fig. 4). The $Sc–P_\alpha$ and $P_\alpha–C49$ bong lengths in **9** (2.544(1) and 1.912(2) Å) are slightly longer than the $Sc–P_\alpha$ and $P_\beta–C57$ bong lengths in **7** (2.508(1) and 1.844(2) Å); however, the $P_\alpha–P_\beta$ length in **9** (2.222 (1) Å) is much longer than that in **7** (2.050(1) Å). Complex **2** also reacts with ethylene at room temperature, but gives a complicated mixture.

**Computational studies.** In order to get some insights on the bonding properties of scandium phosphinophosphinidene complex, DFT calculations on complex **1** were carried out. Scrutinizing the molecular orbitals indicates that the HOMO-1 and

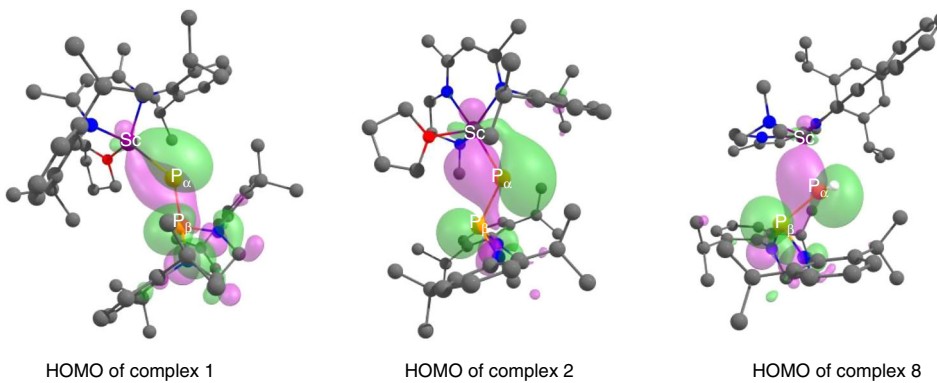

**Fig. 5 HOMO of complexes 1, 2, and 8.** The HOMO presents a donor–acceptor interaction between the lone pair on $P_\beta$ and Sc. Atom color code: purple, scandium; orange, phosphorus; red, oxygen; blue, nitrogen; gray, carbon; and white, hydrogen.

HOMO-2 are Sc–$P_\alpha$ π and σ bonding interactions, respectively, (Supplementary Fig. 49), whereas the HOMO displays a donor–acceptor interaction between the lone pair on $P_\beta$ and Sc (Fig. 5). This bonding situation is further confirmed by NBO analysis. The Wiberg Indexes are 1.4 and 0.3 for the Sc–$P_\alpha$ and Sc–$P_\beta$ bonds, respectively, in line with a double bond character for the former and a donor–acceptor nature for the latter (for set of comparison the P–P bond has a Wiberg Index of 1.0).

For the sake of comparison, the bonding situation in complex **2** was also investigated computationally. First the molecular orbitals were analyzed. Even though HOMO-1 and HOMO-2 clearly shows σ + π interaction between Sc and $P_\alpha$ in line with a Sc=$P_\alpha$ double bond character (Supplementary Fig. 50), the HOMO (Fig. 5) is quite different from that of **1** since the donation from the $P_\beta$ to Sc seems weaker. This is further highlighted by the NBO analysis and the Wiberg indexes. Indeed, the Wiberg Indexes are 1.3 and 0.2 for the Sc–$P_\alpha$ and Sc–$P_\beta$ bonds, and 0.7 for the P–P bond, in line with a weaker interaction between the phosphino-phosphinidene and metal center. This difference of bonding in **1** and **2** would affect the reactivity, as in **1** the stronger metal–ligand interaction seems to take place.

To rationalize the peculiar reactivity of complex **1**, the Fukui condensed descriptors of **1** were computed and indicate that $P_\alpha$ has to be a better nucleophile than $P_\beta$ ($\Delta f$ being −0.19 and −0.09, respectively) (Supplementary Table 2), which is not in line with the observed reactivity and therefore, the reaction profile was determined in the gas phase at room temperature (Fig. 6). The reaction begins with the formation of a phenylacetylene adduct whose formation is endothermic by 23.9 kcal mol$^{-1}$. Interestingly, the computed full dissociation energy of THF requires 15.7 kcal mol$^{-1}$. Therefore, the phenylacetylene coordination is endothermic by 8.2 kcal mol$^{-1}$ when the THF is just displaced from the coordination sphere but remain in the vicinity of the metal complex. This makes complicated any reaction of the substrate on the Sc–$P_\alpha$ bond, such as proton transfer or 1,2 addition, that would occur on the same side as THF, that needs to be removed prior to any substrate coordination (Supplementary Figs. 52 and 53) but allows reactivity on the Sc–$P_\beta$ bond (Fig. 6). From the phenylacetylene adduct, the cycloaddition occurs very easily with a barrier of 27.0 kcal mol$^{-1}$ (3.1 kcal mol$^{-1}$ from the adduct). Following the intrinsic reaction coordinate leads to the final product **6**, whose formation is exothermic by 3.1 kcal mol$^{-1}$. The kinetic study on the reaction of **1** with phenylacetylene was carried out (see the Supplementary Methods). The reactions of **1** with one equiv. of 1-phenylpropyne in the presence of 75 equiv of THF in toluene-$d_8$ at six different temperatures between −18 and 2 °C were monitored by $^1$H NMR spectroscopy, and an Eyring analysis provided activation parameters of $\Delta H^\ddagger = 15.0(4)$ kcal

mol$^{-1}$, $\Delta S^\ddagger = 0(1)$ cal mol$^{-1}$ K$^{-1}$ and $\Delta G^\ddagger = 15.0$ kcal mol$^{-1}$ for 298.15 K. According to the reaction condition, these activation parameters relate to the THF exchange and compare well with the computed one (15.7 kcal mol$^{-1}$). To further probe the importance of the THF coordination in the reactivity, the coordination of N-benzylidenepropylamine to **1** has been computed. The THF replacement is found to be thermodynamically favored by 8.8 kcal mol$^{-1}$, so that reaction can occur on the most nucleophilic phosphorus in line with the experimental observation. This favorable coordination of N-benzylidenepropylamine is easily explained by the presence of the nitrogen lone pair that ensures the coordination to the metal center. Moreover, the latter also prevents any hydrogen transfer to the phosphinophosphinidene ligand. Interestingly, the [2 + 2] product formation is thermodynamically favorable (−34.9 kcal mol$^{-1}$), whereas the [2 + 3] is disfavored (13.0 kcal mol$^{-1}$) in line with the nucleophilicity.

The somewhat normal reactivity of complex **2** with phenylacetylene was also investigated computationally at the same level of theory (DFT, B3PW91). The THF dissociation energy from complex **2** was computed and found to be 3.4 kcal mol$^{-1}$ only (Fig. 6), which is four times lower than that in complex **1**. This is due to the presence of the donation from the labile amino group on the diketiminato ligand, and in line with the longer Sc–O (THF) bond length found by XRD in complex **2** (2.275(2) vs 2.197(2) Å in complex **1**). Moreover, this also explains the difference of reactivity, since with such an easy dissociation of THF the reactivity at the strongly nucleophilic $P_\alpha$ (NPA charge of −0.23 vs +0.69 at $P_\beta$) can occur. Interestingly, the 1,2 insertion of phenylacetylene on the Sc–$P_\beta$ was found at a similar barrier as found for **1** (20.8 kcal mol$^{-1}$, see Supplementary Fig. 54), which is not competitive with the formation of complex **8** (Fig. 6). Indeed, the THF to phenylacetylene replacement occurs at low energy (5.6 kcal mol$^{-1}$, meaning 2.2 kcal mol$^{-1}$ with respect to the THF dissociation). From there, the proton transfer transition state was located and the associated barrier is very low (7.2 kcal mol$^{-1}$ with respect to the entrance channel that is 1.6 kcal mol$^{-1}$ only after the THF replacement) in line with a very rapid reaction, as observed experimentally. This is in line with the high nucleophilic character of the $P_\alpha$ that abstracts the proton of phenylacetylene. Replacing the hydrogen by a methyl group in the substrate would prevent this reaction and therefore yield a [2 + 2] addition product. This is exactly what is observed experimentally. Following the intrinsic reaction coordinate yields the formation of the phenylacetylide complex **8**, whose formation is strongly exothermic (−15.6 kcal mol$^{-1}$).

The bonding situation in **8** was analyzed in order to compare with that in **2**. In terms of molecular orbitals, the HOMO exhibits a $P_\beta$ to Sc donation (Fig. 5), whereas only the HOMO-1 indicates

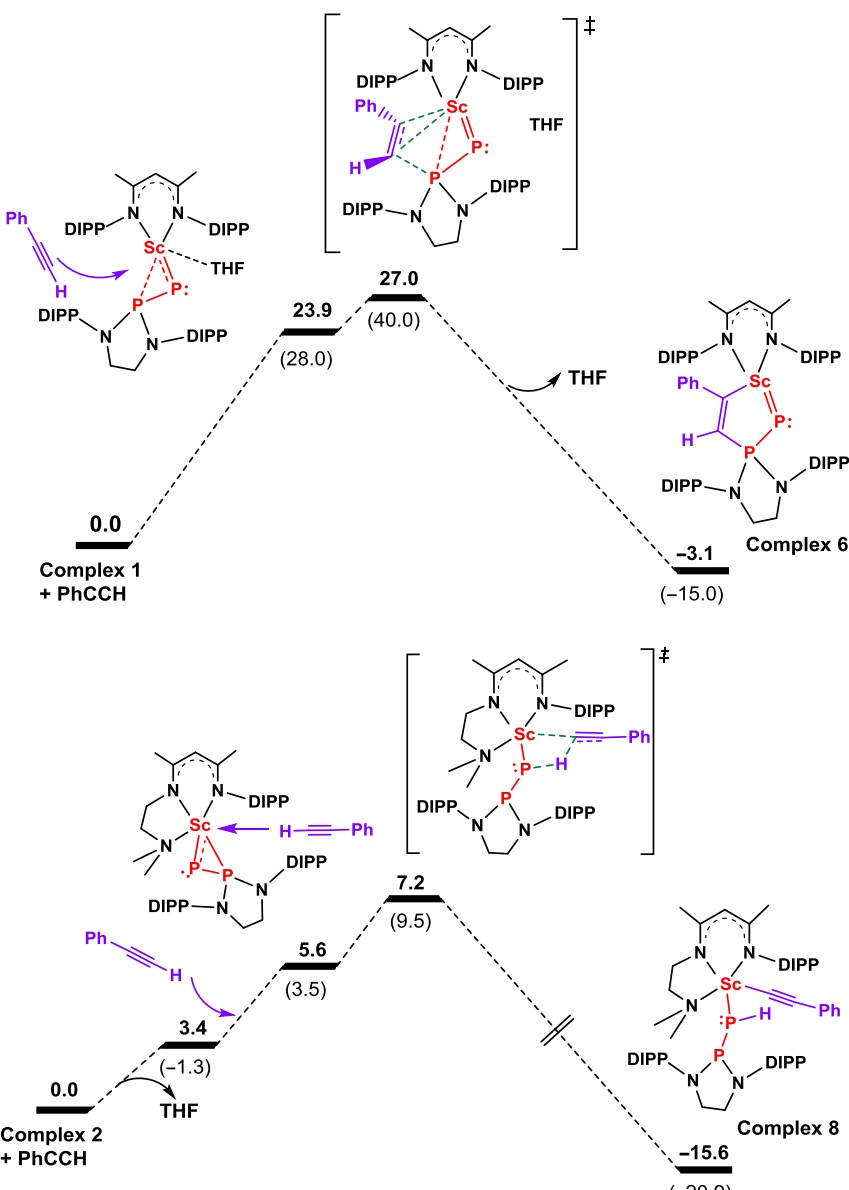

**Fig. 6 Computed enthalpy profiles (in kcal mol⁻¹).** The reactions of phenylacetylene with complexes **1** and **2** provide complexes **6** and **8**, respectively. The values in brackets are the Gibbs free energy.

a σ Sc–$P_\alpha$ bond (Supplementary Fig. 55). This is further corroborated by NBO and Wiberg bond indexes analysis. Indeed, the Wiberg Indexes are 0.9 and 0.1 for the Sc–$P_\alpha$ and Sc–$P_\beta$ bonds, respectively, and 0.6 for the P–P bond (for sake of comparison the P–H Wiberg Index is 0.1). Therefore, the phosphide ligand is thus described as a Sc–$P_\alpha$ single bond with a donation from the lone pair of $P_\beta$.

## Discussion

In this contribution, we report the synthesis, structures, and reactivity of two scandium phosphiniophosphinidene complexes (**1** and **2**). These two complexes are prepared from reactions of potassium salt K[HPP{N(DIPP)CH₂CH₂N(DIPP)}] with scandium methyl chloride [LSc(Me)Cl] or [L′Sc(Me)Cl] in a THF/toluene mixture via salt elimination and subsequent methane elimination. Complexes **1** and **2** are monometallic, and the bonding analysis on the complexes clearly indicates the presence of a Sc=$P_\alpha$ double bond and a weak Sc–$P_\beta$ donor–acceptor interaction. Complex **1** undergoes a 1,3 addition with 2,2′-bipyridine, ethylene, phenylacetylene, or

1-phenyl-1-propyne (reaction at the least nucleophilic site $P_\beta$), showing a peculiar reactivity. Meanwhile, complex **2** presents a normal reactivity at the most nucleophilic phosphorus site ($P_\alpha$), such as a 1,2 addition with 1-phenyl-1-propyne and a proton transfer with phenylacetylene. This intriguing difference of reactivity is rationalized using DFT calculations, which demonstrate that the abnormal reactivity of complex **1** is induced by the strong coordination of the THF molecule in complex **1** preventing the reactivity at $P_\alpha$. Therefore, the work demonstrates that the rare-earth monometallic phosphinidene complex is feasible, and reveals a preliminary reactivity of the Sc=P double bond (complex **2**). This work creates a new horizon for rare-earth metal chemistry and metal–ligand multiple bonding chemistry.

## Methods

**General considerations**. Experiments were carried out under an atmosphere of argon using Schlenk techniques or in a nitrogen filled glovebox. All solvents and reagents were rigorously dried and deoxygenated before use. All the new compounds were characterized by NMR and single crystal XRD, the new compounds except **2** were also characterized by elemental analyses. A satisfied elemental analysis for **2** was

not obtained as the compound decomposed when it was dried under vacuum. Calculations were carried out with Gaussian09 at the DFT level, with the hybrid functional B3PW91. The synthetic procedures of H$_2$PP{N(DIPP)CH$_2$CH$_2$N(DIPP)} and **1** are listed as below, their NMR and elemental analyses data are also listed. The synthetic procedures and characterization of **2–9** are generally similar to those of **1**, and are provided in Supplementary Methods. See Supplementary Methods for the single crystal XRD analysis, the kinetic study and the theoretical calculations; see Supplementary Data for the datasets of Cartesian coordinates in calculations.

**Synthesis of H$_2$PP{N(DIPP)CH$_2$CH$_2$N(DIPP)}.** ClP{N(DIPP)CH$_2$CH$_2$N(DIPP)} (667 mg, 1.5 mmol) and NaPH$_2$ (84 mg, 1.5 mmol) were mixed in 7 mL of THF. After stirring at room temperature for 4 h, the volatiles of the solution were removed under vacuum, and the residue was extracted with hexane (15 mL). The resulting solution was concentrated to 2 mL, and then cooled to −35 °C to give a yellow crystalline solid. The solid was collected and dried under vacuum to give H$_2$PP{N(DIPP)CH$_2$CH$_2$N(DIPP)} as a yellow solid (264 mg, 40% yield). $^1$H NMR (400 MHz, C$_6$D$_6$, 25 °C): δ (ppm) 7.25–7.06 (m, Ar*H* of DIPP, overlapped with the residual solvent resonance of the deuterated solvent), 3.82–3.64 (m, 6H, NC*H*$_2$ and C*H*Me$_2$), 3.20 (m, 2H, NC*H*$_2$), 2.22 (dd, $^1J_{P–H}$ = 183.1 Hz, $^2J_{P–H}$ = 15.7 Hz, 2H, P*H*$_2$), 1.38 (d, $^3J_{H–H}$ = 6.8 Hz, 6H, CH*Me*$_2$), 1.29 (d, $^3J_{H–H}$ = 6.8 Hz, 6H, CH*Me*$_2$), 1.26 (d, $^3J_{H–H}$ = 6.8 Hz, 6H, CH*Me*$_2$), 1.24 (d, $^3J_{H–H}$ = 6.8 Hz, 6H, CH*Me*$_2$). $^{13}$C {$^1$H} NMR (100 MHz, C$_6$D$_6$, 25 °C): δ (ppm) 150.0 (d, $^3J_{P–C}$ = 2.6 Hz, *o*-Ar*C* of DIPP), 148.8 (*o*-Ar*C* of DIPP), 137.9 (d, $^2J_{P–C}$ = 13.1 Hz, *i*-Ar*C* of DIPP), 124.8 (*m*-Ar*C* of DIPP), 124.7 (*p*-Ar*C* of DIPP), 54.9 (d, $^2J_{P–C}$ = 8.0 Hz, NC*H*$_2$), 29.2, 29.1, 29.04, 28.97 (*C*HMe$_2$), 25.8, 24.54, 24.52, 24.19, 24.16 (CH*Me*$_2$). $^{31}$P NMR (162 MHz, C$_6$D$_6$, 25 °C): δ (ppm) 134.9 (dt, $^1J_{P–P}$ = 186.0 Hz, $^2J_{P–H}$ = 15.7 Hz, P$_\beta$), −157.6 (dt, $^1J_{P–P}$ = 186.0 Hz, $^1J_{P–H}$ = 183.2 Hz, P$_\alpha$). Anal. Calcd for C$_{26}$H$_{40}$N$_2$P$_2$: C 70.56; H 9.11; N 6.33. Found: C 70.88; H 9.19; N 6.27.

**Synthesis of 1.** KCH$_2$(C$_6$H$_5$) (33 mg, 0.25 mmol) was added to a THF solution (2 mL) of H$_2$PP{N(DIPP)CH$_2$CH$_2$N(DIPP)} (111 mg, 0.25 mmol) at −35 °C. After standing at −35 °C overnight, to the reaction solution was added a toluene suspension (4 mL) of [LSc(Me)Cl] (128 mg, 0.25 mmol). After standing at room temperature for 24 h, the solvent was removed under vacuum and the residue was extracted with toluene (8 mL). The solvent of the extraction was removed under vacuum, the residue was washed with hexane (1 mL) and dried under vacuum to give **1**·hexane (complex **1** was obtained with hexane in the lattice) as a dark red solid (143 mg, 54% yield). $^1$H NMR (400 MHz, C$_6$D$_6$, 25 °C): δ (ppm) 7.31–6.96 (m, Ar*H* of DIPP, overlapped with the residual solvent resonance of the deuterated solvent), 6.87 (m, 2H, Ar*H* of DIPP), 4.64 (s, 1H, MeC(N)C*H*), 4.35 (sept, $^3J_{H–H}$ = 6.8 Hz, 2H, C*H*Me$_2$), 4.22 (sept, $^3J_{H–H}$ = 6.8 Hz, 2H, C*H*Me$_2$), 3.94 (m, 8H, NC*H*$_2$, C*H*Me$_2$ and THF-*H*), 3.35 (m, 2H, NC*H*$_2$), 2.21 (sept, $^3J_{H–H}$ = 6.8 Hz, 2H, C*H*Me$_2$), 1.90 (d, $^3J_{H–H}$ = 6.8 Hz, 6H, CH*Me*$_2$), 1.39–1.20 (m, 36H, CH*Me*$_2$, THF-*H*, C*Me* and hexane-*H*), 1.13 (d, $^3J_{H–H}$ = 6.8 Hz, 6H, CH*Me*$_2$), 0.98 (d, $^3J_{H–H}$ = 6.8 Hz, 6H, CH*Me*$_2$), 0.89 (t, $^3J_{H–H}$ = 6.4 Hz, hexane-*H*), 0.49 (d, $^3J_{H–H}$ = 6.8 Hz, 6H, CH*Me*$_2$), 0.41 (d, $^3J_{H–H}$ = 6.8 Hz, 6H, CH*Me*$_2$). The solubility of **1**·hexane in C$_6$D$_6$ is low, therefore its $^{13}$C{$^1$H} NMR spectrum was recorded in THF-*d*$_8$. $^{13}$C{$^1$H} NMR (100 MHz, THF-*d*$_8$, 25 °C): δ (ppm) 168.4 (imine *C*), 152.7, 149.9, 146.1, 143.3 (Ar*C* of DIPP), 143.1 (d, $^2J_{P–C}$ = 8.0 Hz, *i*-Ar*C* of DIPP), 141.8, 126.4, 126.3, 125.5, 124.9, 123.3, 123.1 (Ar*C* of DIPP), 98.4 (MeC(N)*C*H), 68.0 (THF-*C*), 54.6 (d, $^2J_{P–C}$ = 8.0 Hz, NC*H*$_2$), 29.7, 29.3, 28.9, 28.6 (*C*HMe$_2$), 28.2, 26.4, 25.9, 25.3, 24.7, 23.4, 23.2 (CH*Me*$_2$), 26.2 (THF-*C*), 24.9 (C*Me*), 32.4, 23.3, 14.3 (hexane-*C*). $^{31}$P {$^1$H} NMR (162 MHz, C$_6$D$_6$, 25 °C): δ (ppm) 412.0 (d, $^1J_{P–P}$ = 504.9 Hz, P$_\alpha$), 157.2 (d, $^1J_{P–P}$ = 504.9 Hz, P$_\beta$). $^{31}$P{$^1$H} NMR (162 MHz, THF-*d*$_8$, 25 °C): δ (ppm) 402.3 (d, $^1J_{P–P}$ = 500.8 Hz, P$_\alpha$), 158.5 (d, $^1J_{P–P}$ = 500.8 Hz, P$_\beta$). Anal. Calcd for C$_{59}$H$_{87}$N$_4$OP$_2$Sc·hexane: C 73.55; H 9.59; N 5.28. Found: C 73.52; H 9.71; N 5.07.

## Data availability

Crystallographic data for the structures reported in this article have been deposited at the Cambridge Crystallographic Data Centre (CCDC) under deposition nos. CCDC 1945178 (H$_2$PP{N(DIPP)CH$_2$CH$_2$N(DIPP)}), 1945179 (**1**), 1958510 (**2**), 1945180 (**3**), 1945181 (**4**), 1945182 (**5**), 1945183 (**6**), 1945184 (**7**), 1958511 (**8**), and 1958512 (**9**). These data can be obtained free of charge from the Cambridge Crystallographic Data Centre via www.ccdc.cam.ac.uk/data_request/cif. All other data supporting the findings of this study are available within the Article and its Supplementary Information, at the Oxford University Research Archive (https://ora.ox.ac.uk) and from the corresponding authors upon reasonable request.

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

## Acknowledgements

This work was supported by the National Natural Science Foundation of China (Nos. 21732007, 21890721, and 21821002), K. C. Wong Education Foundation, the Program of Shanghai Academic Research Leader, the Strategic Priority Research Program of the Chinese Academy of Sciences (Grant No. XDB20000000), and Fujian Institute of Innovation, Chinese Academy of Sciences. L.M. acknowledges the Chinese Academy of Sciences President's International Fellowship Initiative.

## Author contributions

Y.C. conceived this project. B.F. performed the synthesis experiments. X.L. solved all of the X-ray structures. Y.C., B.F., and L.X. analyzed the experimental data. K.N.M. and L.M. conducted the theoretical computations and analyzed the results. Y.C. and L.M. drafted the paper with support from B.F., L.X., and K. M. All the authors discussed the results and contributed to the preparation of the final manuscript.

## Competing interests

The authors declare no competing interests.
