## [Peer Review File · Nature Communications]

Reviewers' comments:

Reviewer #1 (Remarks to the Author):

This paper describes the interesting reactivity of a Scandium phosphinophosphinidene complexes with a series of small molecules. While the complexes are interesting and unique, the chemistry results from the presence of a Lewis acidic Sc and the basic P center which do not interact directly. Indeed this is directly analogous to frustrated Lewis pair chemistry. Additions to alkyne, olefin, imine, pyridines, as well as deprotonation of alkyne have been seen with FLPs. Thus I was surprised that the authors did not draw this analogy. In particular it should be pointed out that Wass has reviewed metal-based FLP chemistry. The present work really should be discussed in this context. There is no doubt that the chemistry reported is well done, thorough and interesting.

Overall, while the reactivity is interesting and the work well done, I am not convinced this is the conceptual advance one would expect for Nature Comm.

Reviewer #2 (Remarks to the Author):

The authors in this manuscript reported synthesis, structural characterization and reactivity studies of two scandium phosphinophosphinidene complexes. The structural studies show that the complexes contain Sc-P multiple bonds, which are very unusual for scandium metal.

The results are certainly interesting and exciting in inorganic chemistry community. I support publication of this work in Nature Communications.

Some minor concerns are as follows.

(1) The DFT studies account for the different reactivities between complexes 1 and 2. However, there is no any discussion on how different unsaturated substrates show very different reaction patterns. Some substrates react via [2+2] while some react via [2+3] or others. Additional calculations and discussion are necessary.

(2) In the energy profiles give, relative enthalpies are given. The authors are better to provide relative Gibbs free energies as well.

Reviewer #3 (Remarks to the Author):

This manuscript describes the synthesis and characterization and reactivity of scandium phosphinophosphinidene complexes supported by beta-diimino ligands. Different from the phosphinidene complexes having a carbonyl substituent at the P atom reported previously, which usually formed a dimeric structure through intermolecular metal-ligand interactions (refs. 17-23), the phosphinophosphinidene complexes reported in this work adopted a monomeric structure because of an intramolecular chelation interaction between the P-P unit and the Sc atom. The P-P chelation with Sc

in complex 1 led to the reactions with ethylene, alkynes and bpy in a 3+2 addition fashion. This is different from what could be expected for a normal phosphinidene rare-earth metal complex. Complex 2 showed the expected normal reactivity, because it has an amino side arm at the beta-diimino ligand, whose coordination to Sc made the Sc-P(beta) interaction (or chelation) weaker. All complexes reported in this paper are well characterized by NMR and X-ray crystallographic analyses. The bonding characters of the phosphinidene complexes 1 and 2 and their reactivity toward phenyl acetylene were also analyzed by the DFT calculations. Overall, this work is well done and offers some new insights into the rare earth phosphinidene chemistry.

The authors may wish to address the following issues.

1. It is concluded that the abnormal reactivity of complex 1 is due to the strong coordination of the THF molecule which could hamper a reaction at $P\alpha$. Is it possible to synthesize THF-free phosphinidene complexes without using THF as a solvent?
2. What can we learn from this work for the synthesis of mononuclear rare earth terminal phosphinidene complexes? The phosphinophosphinidene species reported here did give a monomeric structure, but not a true terminal phosphinidene.
3. A tabular summary of important bond lengths/angles, including those reported previously if possible, is highly desired to improve the readability of structure comparison.
4. Typo: Page 6, "In 2 (Fig. 4)".

Reviewers' comments and our response

Reviewer #1 (Remarks to the Author):

This paper describes the interesting reactivity of a Scandium phosphinophosphinidene complexes with a series of small molecules. While the complexes are interesting and unique, the chemistry results from the presence of a Lewis acidic Sc and the basic P center which do not interact directly. Indeed this is directly analogous to frustrated Lewis pair chemistry. Additions to alkyne, olefin, imine, pyridines, as well as deprotonation of alkyne have been seen with FLPs. Thus I was surprised that the authors did not draw this analogy. In particular it should be pointed out that Wass has reviewed metal-based FLP chemistry. The present work really should be discussed in this context. There is no doubt that the chemistry reported is well done, thorough and interesting.

Overall, while the reactivity is interesting and the work well done, I am not convinced this is the conceptual advance one would expect for Nature Comm.

Response: We appreciate the reviewer's suggestion that a brief introduction of metal-based FLP chemistry will be benefit for understanding the reactivity of the P_{β} atom in complex 1. In the revised manuscript, the following sentence has been added. "It's worthy to note that the reactions occur on the Sc- P_{β} bond of **1** are similar to those observed for some metal-based frustrated Lewis pairs, where the unsaturated substrates inserted into the metal-phosphorus functions." The corresponding references have also been added.

However, the most significant part of this paper is the synthesis of the scandium phosphinophosphinidene complex, which represents the first example of rare-earth monometallic phosphinidene complex. The rare-earth monometallic phosphinidene complex is very synthetically challenging as it very easily undergoes ligand redistribution to yield rare-earth bi- or trimetallic phosphinidene complex and lose metal-phosphorus multiple bond. By using a well-designed phosphinidene ligand, we finally synthesized this type of complexes. The use of complex 2 allowed us to demonstrate the expected reactivity of the Sc- P_{α} multiple bond.

However, the reactivity of complex 1 is peculiar since even though P_{α} atom is more nucleophilic than P_{β} atom, reactions with 2,2'-bipyridine, ethylene, phenylacetylene and 1-phenyl-1-propyne occur at the least nucleophilic site P_{β} . Therefore, we carried out the DFT studies to explain such reactivity.

Reviewer #2 (Remarks to the Author):

The authors in this manuscript reported synthesis, structural characterization and reactivity studies of two scandium phosphinophosphinidene complexes. The structural studies show that the complexes contain Sc-P multiple bonds, which are very unusual for scandium metal.

The results are certainly interesting and exciting in inorganic chemistry community. I support publication of this work in Nature Communications.

Some minor concerns are as follows.

(1) The DFT studies account for the different reactivities between complexes 1 and 2. However, there is no any discussion on how different unsaturated substrates show very different reaction patterns. Some substrates react via [2+2] while some react via [2+3] or others. Additional calculations and discussion are necessary.

Response: We thank the reviewer for this suggestion. A calculation was undertaken on the reaction of complex 1 with N-benzylidenpropylamine where a [2+2] addition is observed instead of the [2+3] one found with phenylacetylene. The coordination of the N-benzylidenpropylamine to 1 is very strong so that the THF displacement is found to be exothermic by 8.8 kcal mol⁻¹. Thus, complex 1 can react like complex 2, that is involving the most nucleophilic phosphorus (P_α)! Indeed, the product of the [2+2] addition is exothermic by 34.0 kcal mol⁻¹ whereas the [2+3] is endothermic by 13.0 kcal mol⁻¹. It should also be noted that the coordination of the substrate through the nitrogen lone pair prevents the hydrogen transfer from C and therefore a cycloaddition is the only possible reactivity. A discussion was included in the main text “To further probe the importance of the THF coordination in the reactivity, the coordination of N-benzylidenpropylamine to 1 has been computed. The THF replacement is found to be thermodynamically favored by 8.8 kcal mol⁻¹, so that reaction can occur on the most nucleophilic phosphorus in line with the experimental observation. This favorable coordination of N-benzylidenpropylamine is easily explained by the presence of the nitrogen lone pair that ensures the coordination to the metal center. Moreover, the latter also prevents any hydrogen transfer to the phosphinophosphinidene ligand. Interestingly, the [2+2] product formation is thermodynamically favorable (-34.9 kcal mol⁻¹) whereas the [2+3] is disfavored (13.0 kcal mol⁻¹) in line with the nucleophilicity.”

In the case of complex 2, the proton transfer rather than a [2+2] cycloaddition is explained by the nucleophilicity of the P_α atom of the phosphinophosphinidene ligand. However, it was maybe not clear enough so that we add the following sentences in the main text “This is in line with the high nucleophilic character of the P_α that abstracts the proton of phenylacetylene. Replacing the hydrogen by a methyl group in the substrate would prevent this reaction and therefore yield a [2+2] addition product. This is exactly what is observed experimentally.”

(2) In the energy profiles given, relative enthalpies are given. The authors are better to provide relative Gibbs free energies as well.

Response: This is now done on the revised version for all profiles. The Gibbs Free Energy are given in the brackets.

Reviewer #3 (Remarks to the Author):

This manuscript describes the synthesis and characterization and reactivity of scandium phosphinophosphinidene complexes supported by beta-diimino ligands. Different from the phosphinidene complexes having a carbyl substituent at the P atom reported previously, which usually formed a dimeric structure through intermolecular metal-ligand interactions (refs. 17-23), the phosphinophosphinidene complexes reported in this work adopted a monomeric

structure because of an intramolecular chelation interaction between the P-P unit and the Sc atom. The P-P chelation with Sc in complex 1 led to the reactions with ethylene, alkynes and bpy in a 3+2 addition fashion. This is different from what could be expected for a normal phosphinidene rare-earth metal complex. Complex 2 showed the expected normal reactivity, because it has an amino side arm at the beta-diimino ligand, whose coordination to Sc made the Sc-P(beta) interaction (or chelation) weaker. All complexes reported in this paper are well characterized by NMR and X-ray crystallographic analyses. The bonding characters of the phosphinidene complexes 1 and 2 and their reactivity toward phenyl acetylene were also analyzed by the DFT calculations. Overall, this work is well done and offers some new insights into the rare earth phosphinidene chemistry.

The authors may wish to address the following issues.

1. It is concluded that the abnormal reactivity of complex 1 is due to the strong coordination of the THF molecule which could hamper a reaction at P α . Is it possible to synthesize THF-free phosphinidene complexes without using THF as a solvent?

Response: We appreciate the reviewer's point. We synthesized the complex 1 in a THF/toluene mixture (1:2 in the volume) but the THF was still present because we used a THF solution of $K[HPP\{N(DIPP)CH_2CH_2N(DIPP)\}]$ as the reactant. We have tried to synthesize the $K[HPP\{N(DIPP)CH_2CH_2N(DIPP)\}]$ in toluene or benzene, but failed (Actually, we did that before we prepared this potassium salt in THF). Even if we synthesized the $K[HPP\{N(DIPP)CH_2CH_2N(DIPP)\}]$ in THF, when the THF was removed after the reaction, the potassium salt decomposed. It seems that the THF is important for the stabilization of $K[HPP\{N(DIPP)CH_2CH_2N(DIPP)\}]$. $K[HPP\{N(DIPP)CH_2CH_2N(DIPP)\}]$ needs to be kept in THF. We have also tried to prepare the THF-free phosphinidene complex by removing the THF of complex 1 under vacuum, but did not succeed.

2. What can we learn from this work for the synthesis of mononuclear rare earth terminal phosphinidene complexes? The phosphinophosphinidene species reported here did give a monomeric structure, but not a true terminal phosphinidene.

Response: Due to the energy mismatching between the frontier orbitals of the rare-earth metal and the ligand atom, the mononuclear rare earth terminal phosphinidene complex is very unstable. Using a substituent which delocalizes the electronic density on the phosphorus atom is benefit for the synthesis and stabilization of the mononuclear rare-earth terminal phosphinidene complex. And this substituent should be bulky in order to prevent the formation of rare-earth bi- or trimetallic phosphinidene complex.

3. A tabular summary of important bond lengths/angles, including those reported previously if possible, is highly desired to improve the readability of structure comparison.

Response: We thank the reviewer for this suggestion. A tabular summary of important bond lengths/angles including those reported previously has been added in the revised manuscript.

4. Typo: Page 6, “In 2 (Fig. 4)”.

Response: We thank the reviewer for noticing this typo. “In 2” has been replaced with “In 3”.

REVIEWERS' COMMENTS:

Reviewer #2 (Remarks to the Author):

The authors have addressed my major concerns.

In Figure 6 (main text) and Figures S2-S4 (SI), there are steps involving ligand substitution. How about the TSs for these ligand substitution? I understand that some of the TSs could be difficult to locate. However, here we are talking about a Nature Comm manuscript, and the authors are expected to give all the relevant details.

Reviewer #2 (Remarks to the Author):

The authors have addressed my major concerns.

In Figure 6 (main text) and Figures 52-54 (SI), there are steps involving ligand substitution. How about the TSs for these ligand substitution? I understand that some of the TSs could be difficult to locate. However, here we are talking about a Nature Comm manuscript, and the authors are expected to give all the relevant details.

Response: For the formation of complex 6, the ligand exchange is involved in the cycloaddition TS. For the formation of complex 8 and for all other computed profiles, the ligand exchange follows a dissociative mechanism, meaning that the THF has to dissociate first to allow the coordination of the alkyne. This is due to the fact that both molecules compete for the same coordination site as explained in the main text. Indeed, this dissociation or lack of dissociation is the key to explain the reactivity difference. For the dissociation of THF, the Potential Energy surface was scanned and the energy is just smoothly increasing to the dissociated product so that no transition state exists, as expected for a dissociative reaction.